

# 3D-Analysis of a non-planispiral ammonoid from the Hunsrück Slate: natural or pathological variation?

Julia Stilkerich[1], Trisha A. Smrecak[2] and Kenneth De Baets[1]

[1] Geozentrum Nordbayern, Friedrich-Alexander Universität Erlangen-Nürnberg, Erlangen, Germany
[2] Department of Geology, Grand Valley State University, Allendale, United States of America

## ABSTRACT

We herein examine the only known non-planispirally coiled early Devonian ammonoid, the holotype specimen of *Ivoites opitzi*, to investigate if the host was encrusted *in vivo* and if these sclerobionts were responsible for the trochospiral coiling observed in this unique specimen. To test if the presence of runner-like sclerobionts infested the historically collected specimen of *Ivoites opitzi* during its life, we used microCT to produce a three-dimensional model of the surface of the specimen. Our results indicate that sclerobionts grew across the outer rim (venter) on both sides of the ammonoid conch at exactly the location where the deviation from the planispiral was recognized, and where subsequent ammonoid growth would likely preclude encrustation. This indicates *in vivo* encrustation of the *I. opitzi* specimen, and represents the earliest documentation of the phenomenon. Further, this suggests that non-planispiral coiling in *I. opitzi* was likely pathologically induced and does not represent natural morphological variation in the species. Despite the observed anomalies in coiling, the specimen reached adulthood and retains important identifying morphological features, suggesting the ammonoid was minimally impacted by encrustation in life. As such, appointing a new type specimen—as suggested by some authors—for the species is not necessary. In addition, we identify the sclerobionts responsible for modifying the coiling of this specimen as hederelloids, a peculiar group of sclerobionts likely related to phoronids. Hederelloids in the Devonian are commonly found encrusting on fossils collected in moderately deep environments within the photic zone and are rarely documented in dysphotic and aphotic samples. This indicates that when the ammonoid was encrusted it lived within the euphotic zone and supports the latest interpretations of the Hunsrück Slate depositional environment in the Bundenbach-Gemünden area.

# INTRODUCTION

Ammonoids are an extinct group of externally-shelled cephalopods that are often used to study biostratigraphy, diversity and evolutionary patterns (*Ritterbush et al., 2014*). The ammonoid shell is typically coiled with touching or overlapping whorls, but some forms—so called heteromorphs—deviate from this shape as their shell is not entirely

Corresponding author
Kenneth De Baets,
kenneth.debaets@gmail.com,
kenneth.debaets@fau.de

coiled and/or is trochospirally coiled (*Landman, Tanabe & Davis, 1996*). Mesozoic heteromorphs have convergently evolved in the Upper Triassic, Middle to Upper Jurassic, and multiple times in the Cretaceous (*Wiedmann, 1969*; *Dietl, 1978*; *Cecca, 1997*). Early ammonoids were loosely coiled and can therefore also be considered heteromorphs from a morphological perspective. However, early ammonoids differ in important ways from Mesozoic heteromorphs as their embryonic shell is also uncoiled (*House, 1996*; *De Baets et al., 2012*; *De Baets et al., 2013b*; *De Baets, Landman & Tanabe, 2015*), and not all types of coiling known from the Mesozoic have been reported from the Paleozoic (e.g., trochospiral coiling is thought to be absent). The sole known possible exception was a specimen of *Ivoites opitzi* from the Hunsrück Slate of Germany, which showed evidence for non-planispiral coiling (*De Baets et al., 2013b*). Originally, the holotype specimen was interpreted to exhibit a transitional morphology in the natural variation from gyroconic to trochospiral coiling as observed in the Jurassic heteromorph *Spiroceras* (*Dietl, 1978*).

However, this specimen is also encrusted by epicoles—"any organism that spent its life attached to or otherwise inhabiting the exterior of any more or less hard object"(*Davis, Klofak & Landman, 1999*). In biology, the term chosen for the encrusting organism implies the relationship it has with its host (e.g., *in vivo,* post-mortem), and a wide variety of terminology has been employed for encrusters in the paleontological literature (see discussion in *Taylor & Wilson, 2002*). A general term for encrusting or boring organism being used with growing popularity is 'sclerobiont' and means "organisms living in or on any kind of hard substrate" (*Taylor & Wilson, 2002*). If these epicoles, or sclerobionts, settled on the ammonoid host shell during its life, they could be called epizoa (*Davis, Klofak & Landman, 1999*; *Klug & Korn, 2001*) and may cause deviations from planispiral coiling (oscillations of the shell around the median plane to trochospiral coiling) to abnormalities in the whorl cross-section when overgrowing the epizoa (*Merkt, 1966*; *Keupp, 1992*; *Hautmann, Ware & Bucher, 2017*; *Checa, Okamoto & Keupp, 2002*) and various other pathologies (*Larson, 2007*; *Keupp, 2012*; *De Baets, Keupp & Klug, 2015*; *Keupp & Hoffmann, 2015*). Cephalopod workers have commonly used the term epicoles to refer to organisms which encrust ammonoids post-mortem (*Davis, Klofak & Landman, 1999*; *Klug & Korn, 2001*; *Rakociński, 2011*; *De Baets, Keupp & Klug, 2015*; *Keupp & Hoffmann, 2015*). Deviations from planispiral coiling in ammonoids have been attributed to sclerobionts in the past (discussed below), yet distinguishing between *in vivo* and post-mortem encrustations is rarely straightforward. In some cases it is impossible to tell if encrustation was *in vivo* or post-mortem, but using various lines of evidence (*Seilacher, 1960*; *Seilacher, 1982*; *Baird, Brett & Frey, 1989*; *Davis, Klofak & Landman, 1999*; *Keupp, 2012*; *De Baets, Keupp & Klug, 2015*; *Keupp & Hoffmann, 2015*) can sometimes elucidate a live-live interaction between host and encruster. The main criteria used by researchers to identify likely cases of *in vivo* colonization of cephalopod shell are (compare *Rakús & Zítt, 1993*; *Davis, Klofak & Landman, 1999*; *Klug & Korn, 2001*; *Luci & Cichowolski, 2014*):

(1) both flanks are encrusted but the apertural region remains free of encrusters
(2) encrusters growth stops precisely at a whorl or are otherwise outpaced by the conch growth of the cephalopod

(3) encrusters show a dominant growth direction consistent with shape and putative life position of the cephalopod shell and may express changes in growth direction as the host life position changes

(4) the cephalopod reacts to its encrusters during growth by developing some kind of behavior that is reflected in the shell (usually nonplanispiral coiling and other deformations).

Cases in which deformation of the shell and/or deviation from the normal planispiral coiling were caused by encrusters provide incontrovertible evidence that the encrusters colonized the shell while the host lived (*Checa, Okamoto & Keupp, 2002*; *Luci & Cichowolski, 2014*). Asymmetrical encrustations during life result in deviations from the planispiral; this has been experimentally demonstrated in gastropods and has been observed in various taxa of ammonoids (*Merkt, 1966*; *Klug & Korn, 2001*; *Keupp, 2012*). Patterns related with Criteria 1 and 3 are the only criteria which can be used to infer *in vivo* encrustation when host growth has already stopped (*Seilacher, 1960*; *Keupp, 2012*), but could potentially also develop in post-mortem sclerobiont attachment during necroplanktonic drift. However, post-mortem drift seems unlikely when ammonoid shells are small (<200 mm: *Wani et al., 2005*; *Rakociński, 2011*). Large, well-preserved or heavily colonized ammonoids were likely also encrusted *in vivo*, because the length of time required for significant encrustation to occur is greater than the length of necroplanktonic drift, even when the additional weight of the sclerobiont is not considered (*Keupp, 2012*). Furthermore, a vertical position resembling the living position of the ammonoid is not always preserved in necroplanktonic drift, and a subhorizontal position can be achieved after *asymmetrical* post-mortem encrustation as a result of added weight (e.g., loosely coiled *Spirula; Donovan, 1989*).

Although post-mortem encrustations of ammonoids on the seafloor are inferred to be common (*Rakociński, 2011*), there are many examples for different organisms settling on the shells of living and fossil cephalopods including foraminifers, bivalves, sponges and corals (*Baird, Brett & Frey, 1989*; *Davis & Mapes, 1999*; *Kröger, Servais & Zhang, 2009*; *Keupp, 2012*; *Wyse Jackson & Key Jr, 2014*). These live-live interactions are not necessarily beneficial for the host or the sclerobiont. Often the cephalopods are disadvantaged, because encrustation increases drag and provides an additional weight burden that the cephalopod must carry, potentially limiting speed and mobility (*Keupp, 2012*). In some cases the encrusters have a disadvantage. As the host cephalopod grows, the encruster may rotate away from their preferred position, losing access to valuable currents for filter feeding, and might eventually be overgrown by the shell in coiled ammonoids (*Hautmann, Ware & Bucher, 2017*; *Meischner, 1968*). However, encrusters largely profit from establishing on a pelagic host. Sessile organisms obtain a pseudoplanktic method of locomotion, providing the potential for greater and more varied nutrition and increased reproductive breadth.

Mobile organisms can potentially use the shell as temporary pasture (*Keupp, 2012*), while sclerobionts can use it as benthic island surrounded by soft and unconsolidated sediment (*Seilacher, 1982*). For pathological reactions in shell form and growth to occur, the sclerobionts must settle on still growing, younger hosts. Sclerobionts that settle on the shell of adult animals that have already reached their final shell size do not induce a

pathological change in the host. In those situations, it is only possible to infer that these sclerobionts encrusted *in vivo* because of their preferential orientation with respect to water currents or the life position of its host (*Seilacher, 1960*; *Seilacher, 1982*; *Keupp, Röper & Seilacher, 1999*; *Kröger, Servais & Zhang, 2009*; *Hauschke, Schöllmann & Keupp, 2011*).

If encrustation happens after the host's death, the organisms can colonize both the exterior and interior of empty shells (*Bartels, Briggs & Brassel, 1998*). Shells which are lying on the seabottom are typically substantially overgrown on one side (the portion above the sediment-water interface) and is usually taken as good evidence for post-mortem encrustation (*Seilacher, 1982*; *Schmid-Röhl & Röhl, 2003*; *Lukeneder, 2008*; *Keupp, 2012*). Encrustation on both sides can potentially also develop in reworked shells and internal moulds; however these typically show a more complex history of encrustation involving multiple generations and a variety of taxa (*Macchioni, 2000*; *Luci & Cichowolski, 2014*; *Luci, Cichowolski & Aguirre-Urreta, 2016*). More importantly, resedimentation typically results in shell breakage and reworked ammonoids (sensu *Fernández-López, 1991*) which differ considerably from normally preserved ammonoids (e.g., abrasional features, differences in infilling and preservation: *Fernández-López & Meléndez, 1994*). Post-mortem encrustation can also be recognized when structures normally believed to be covered with soft-parts (inside of the shell) or additional objects are encrusted by the epicoles (*Bartels, Briggs & Brassel, 1998*; *Klug & Korn, 2001*). Different generations of sclerobionts with clearly diverging orientations or different taxa on both sides of the ammonoid are also indicative of a post-mortem encrustation (*Macchioni, 2000*; *Klug & Korn, 2001*; *Luci & Cichowolski, 2014*).

Our main goal is to test if the sclerobionts settled on the ammonoid during their lifetime, which can tested by investigating the criteria listed above—particularly if they are growing on both sides of the shell (criterium 1) and if the beginning of non-planispiral coiling (criterium 4) correlates with the settling of these sclerobionts. If these encrustations happened during their lifetime and can be linked with severe pathological reactions (e.g., non-planispiral coiling), this might have important implications for taxonomy and indirectly for biostratigraphy (*Spath, 1945*). Pathological specimens with strongly different morphologies have occasionally been described as different species (*Spath, 1945*; *Keupp, 2012*).

An additional goal is to identify the identity of the sclerobionts, which were preliminarily determined to be auloporid tabulate corals (*De Baets et al., 2013b*). Some taxa of auloporid corals have traditionally been confused with other sclerobionts with runner-like morphologies (*Lescinsky, 2001*) like hederelloids and cyclostomate bryozoans (*Fenton & Fenton, 1937*; *Elias, 1944*; *Bancroft, 1986*).

In testing these questions, it was important to avoid using destructive analyses because the specimen is an important historical specimen (*Opitz, 1932*) and the holotype of *Ivoites opitzi* (*De Baets et al., 2013b*) from the famous Hunsrück Lagerstätte. The Hunsrück Slate is a facies typical for the Lower Devonian (Emsian) of the Rhenish Massif which consists predominantly of dark fine-grained argillites metamorphosed into slates (*Bartels, Briggs & Brassel, 1998*). In the Bundenbach-Gemünden area, these strata can contain fossils with remarkable preservation including articulated echinoderms and vertebrates as well

as preserved soft tissues of arthropods and other groups without hard tissues (*Bartels, Briggs & Brassel, 1998*). Although some fossils reveal remarkable preservation, they are all typically flattened and it is difficult to impossible to prepare such thin, compressed fossils from both sides without destroying parts of it. This is for example illustrated by the only known specimen and holotype of *Palaeoscorpius devonicus*, where some parts of the shale that are thinner than 1 mm are very fragile or missing altogether after preparation (*Kühl et al., 2012b*). This might be one of the reasons why fossils with hard parts commonly studied for biostratigraphic or paleoenvironmental purposes at other sites where they are more three-dimensionally preserved have been comparably little studied in the Hunsrück Slate (*Bartels, Briggs & Brassel, 1998*; *Südkamp, 2007*). This is also the case for ammonoids, which are important index fossils to date this deposit and are often extremely flattened hampering also their taxonomic assignment (*Bartels, Briggs & Brassel, 1998*; *De Baets et al., 2013a*; *De Baets et al., 2013b*; *Übelacker, Jansen & De Baets, 2016*).

Considering the size and the preservation of our specimen, as well as the expected X-ray contrast between pyritic fossils and the slate matrix, we elected micro-CT to create a three-dimensional model to answer these questions. This method is well suited for these purposes (*Sutton, Rahman & Garwood, 2014*). Many CT-studies have focused on analyzing ontogeny or morphological traits for phylogenetic purposes (*Monnet et al., 2009*; *Garwood & Dunlop, 2014*; *Naglik et al., 2015a*), but they can be used to test ecological or paleobiological aspects (*Kruta et al., 2011*; *Kühl et al., 2012b*; *Hoffmann et al., 2014*; *Takeda et al., 2016*) such as the interpretation of pathologies (*Anné et al., 2015*) and bioerosion (*Beuck et al., 2008*; *Rahman et al., 2015*). Tomographic studies in ammonoids have particularly focused on functional morphology, empirical buoyancy calculations and ontogeny of the chambered shell (*Lukeneder, 2012*; *Hoffmann et al., 2014*; *Tajika et al., 2014*; *Lemanis et al., 2015*; *Naglik et al., 2015a*; *Naglik et al., 2015b*; *Tajika et al., 2015*; *Lemanis et al., 2016*; *Lemanis, Zachow & Hoffmann, 2016*; *Naglik, Rikhtegar & Klug, 2016*).

## MATERIAL AND GEOLOGICAL SETTING

The studied fossil specimen is the holotype of *Ivoites opitzi*, which was collected from the Hunsrück Slate in the Central Hunsrück, now known as the Middle Kaub Formation (*Schindler et al., 2002*), at the Schieleberg-quarry near Herrstein, Germany (*De Baets et al., 2013b*; see Fig. 1 for a map and stratigraphic provenance of this specimen). It is reposited in the Karl-Geib-Museum in Bad Kreuznach: KGM 1983/147. The Middle Kaub Formation contains some of most completely preserved early ammonoids (*De Baets et al., 2013b*) and belong the oldest known ammonoid faunas (*Becker & House, 1994*) together with similar aged faunas from China (*Ruan, 1981*; *Ruan, 1996*) and Morocco (*De Baets, Klug & Monnet, 2013*; *De Baets, Klug & Plusquellec, 2010*). The exact stratigraphic position of our specimen is not known. However, *Ivoites* is restricted to Early Emsian. This particular species (*I. opitzi*) has been found associated with dacryoconarid *Nowakia praecursor* in samples deriving from Eschenbach-Bocksberg Quarry, but they have also been found in overlying layers of the Obereschenbach quarry (Wingertshell member sensu *Schindler et al., 2002*), which might range into the Barrandei Zone (*De Baets et al., 2013b*). Other ammonoids,
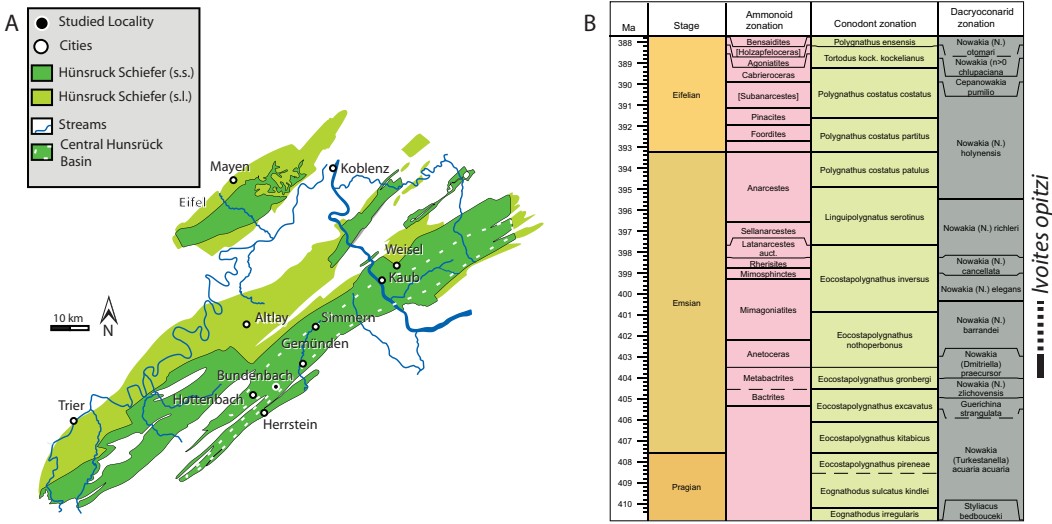

**Figure 1** **Geographic and Stratigraphic Context.** (A) map of the locality (modified from *De Baets, 2012*) and (B) stratigraphic provenance of *Ivoites opitzi* (time-scale based on *Becker, Gradstein & Hammer (2012)*; created with time-scale creator 6.4: http://engineering.purdue.edu/Stratigraphy/tscreator/).

including those from the same genus (*I. schindewolfi*, *Erbenoceras solitarium*), which have been reported from the early Emsian strata of the Schieleberg quarry in Herrstein, speak for a Praecursor to the Barrandei zone age of the strata (*De Baets et al., 2013b*).

We investigated the taxonomy, provenance and taphonomy of 342 ammonoids from the Central Hunsrück Basin in a recent monograph (*De Baets et al., 2013b*) including 82 specimens of *Ivoites* and 7 additional specimens of closely related *Metabactrites*.

The investigated specimen was chosen as the holotype above 19 other specimens of *Ivoites opitzi* as it was the most complete, three-dimensional and well-preserved specimen of the species. It has three complete whorls and a diameter of 105 mm (Fig. 2; *De Baets et al., 2013b*). The specimen is mostly preserved as an internal, pyritic mould as evidenced by traces of suture lines and other structures (e.g., opitzian pits) internal to the shell (taphonomic category IIB of *De Baets et al., 2013b*). The last half-whorl probably corresponds to the body chamber as indicated by faint traces of a suture in hand piece and X-ray images (*Kneidl, 1984*), lack of pyrite infilling of the last whorl, terminal uncoiling as well as the large lateral extension of the end of the whorl interpreted to be the apertural edge (*De Baets et al., 2013b*).

The infilling of the shell with pyrite in this taphonomic category is interpreted to have happened early in the diagenesis, below the sediment-water interface and before the dissolution, compaction, and breakage of the shell. These observations were used successfully to interpret preservation of ammonoids in the Jurassic bioturbated shales (*Hudson, 1982*), and were additionally supported by fracture patterns (*De Baets et al., 2013b*).

The whorls of this specimen touch and overlap each other, but this is interpreted to be a consequence of compression and tectonic deformation as the inner whorl lies completely above the following whorl (*De Baets et al., 2013b*). Oblique embedding can result in one bit

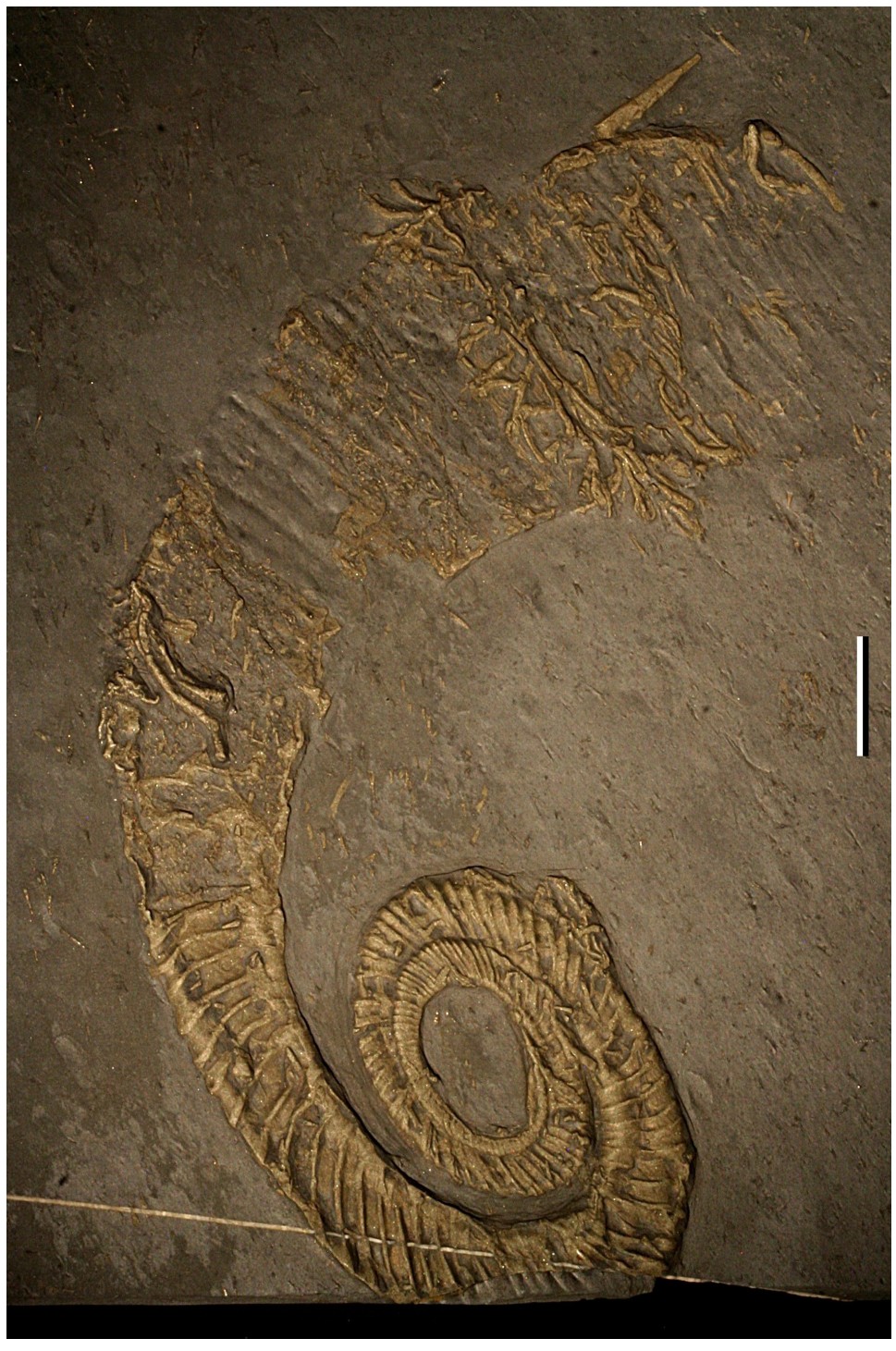

**Figure 2 Holotype of *Ivoites opitzi*.** Several sclerobionts can be seen encrusting specimen KGM 1983/147. Scale bar: 1 cm.
of whorl lying on top of one whorl, but the opposite side should then lie below this whorl (see for example *De Baets et al., 2013b*, Plate 5, Fig. 3 for an example), which is not the case in our specimen. Compaction on an umbilical concretion can also result in asymmetric deformation, but affect similar quadrants in the same way and the concretion should still be present, which is not the case in our specimen (see *De Baets et al., 2013b*, Plate 10, Fig. 11 for an example). Compaction of a horizontally embedded specimen would result in subsequent whorl (with a thicker whorl section) to lie above previous whorls. The only way the preservation observed in KGM 1983/147 could occur is if the specimen was already slightly torticonic before burial.

During the initial investigation of the holotype of *I. opitzi,* the possibility of a live-live relationship between the host and the sclerobionts was excluded because of an inferred lack of conclusive evidence (*De Baets et al., 2013b*). The sclerobionts nearer the aperture did not provide any evidence that could support *in situ* or post-mortem encrustation. The sclerobiont colony on the phragmacone near the point of non-planispiral coiling was seemingly not growing across both sides of the whorl, and again, the authors were unable to infer the relationship between host and sclerobiont. But only the left side was prepared in this historically collected material (*Opitz, 1932*, p. 121, Fig. 117). A microCT makes it possible to study the specimen from both sides, giving us the unique opportunity to reevaluate this interpretation.

## METHODS

The specimen of *I. opitzi* under investigation is a holotype, and thus could not be investigated destructively. Thus, the specimen was well-suited to be studied with X-ray microtomography. X-ray microtomography uses X-rays to acquire radiographs (or projections) of an object at multiple angles. From these projections, a sequence of parallel and evenly spaced tomograms (individual slice images mapping the X-ray attenuation within a sample) is computed indirectly. This tomographic dataset in turn can be used to recreate a virtual model (3D model) without destroying the original object. The prefix micro- refers to the fact that pixel sizes of the cross-sections are in the micrometre range (*Sutton, Rahman & Garwood, 2014*). Michael Wuttke kindly borrowed the specimen from the Karl-Geib-Museum and Markus Poschmann (Koblenz) it to the Steinmann Institute, where it was investigated with X-ray micro-tomography. KGM 1983/147 was scanned on a Phoenix v|tome|x s at 150 kV and 160 μA with 0.400 s of exposure time by Alexandra Bergmann (Steinmann Institute). This resulted in two thousand two hundred unfiltered projections providing a voxel size of 118 μm ($\sim$ pixel size of 118 um). Three-dimensional reconstructions and an animation were produced using the 107 images (tomograms) in the x–z-plane (by Julia Stilkerich using the free software SPIERS (*Sutton, Gardwood & Siveter, 2012*; http://spiers-software.org)). A video as well as files essential for verification can be found in the supplementary material: the latter includes the used image stack and a scansheet with a description of scan setting and specimen information (*Davies et al., 2017*). The fixed threshold value was manually chosen to maximally separate pyritic fossils from the shale matrix, because the objects of focus (ammonoid and epicoles) are

pyritic (see material and methods). Regions of interest were defined using the masking system in SPIERS, allowing them to be rendered separately to have the most conservative interpretation of the position of the pyritic ammonoid vs. epicoles (*Sutton, Rahman & Garwood, 2014*). Colored masks were used in the final representation to distinguish the various features captured: ammonoid (yellow), runner-like epicoles (green), orthoconic nautiloid (red), brachiopod (blue) and dacryoconarids (yellow). This model was imported in Blender 7.28 and enlarged 200% on the *z*-axis to measure the deviation from the planispiral.

## RESULTS

### Position of the epicoles and its relationship with non-planispiral coiling

In the model, the first whorl lies on top of the second whorl. The median plane of the first whorl seems to lie between 1 and 2 mm above that of the second whorl in the model, which must have been even greater before compaction (see discussion). The 3D-model (see Figs. 3 and 4) therefore substantiates the previously hypothesized suspicion (*De Baets et al., 2013b*) that the specimen is not entirely coiled planispirally. Five clusters (A-E) of colonial sclerobionts can be recognized in the 3D-model (see arrows in Figs. 3 and 4).

At least three clusters (C–E) can be recognized on the phragmocone. Additional clusters (A, B) can be found on the final demi-whorl. In the inner whorls, the sclerobionts are located ventrally on both sides of the whorl cross-section (see Figs. 3C–3E, 4C–4E). Their direction of growth and budding follows the spiral axis of the ammonoid shell. The earliest recognizable sclerobionts with respect to the growth direction of the ammonoid (clusters D–E) coincide with the position where non-planispirality can be first recognized (Fig. 5).

More importantly, there is evidence that clusters (C, D and E) are growing on both sides of the ammonoid (Fig. 6). The last demi-whorl of the host ammonoid was not infilled with pyrite, therefore the growth patterns of the sclerobionts in clusters A and B cannot be established with certainty.

Elongated components like the dacryoconarids present in the substrate (marked in yellow in Fig. 7) along with the *I. opitzi* specimen are often orientated along the direction of the paleo-current (*Hladil, Čejchan & Beroušek, 1991*; *Hladil et al., 2014*; *Gügel et al., 2017*). Neither the dacryoconarids nor the epicoles show a preferential orientation with respect to the substrate. Sclerobiont clusters C, D, and E do show a preferential orientation with respect to the spiral axis of the ammonoid shell. Unnamed, small, and bulky components visible in the matrix are probably pyritic nodules of different sizes.

### Morphology of the runner-like sclerobionts

The mode of preservation of the *I. opitzi* specimen, pyritization and internal mould preservation, makes examination of fine details or microstructure of the sclerobionts impossible. Yet the microCT permits three dimensional examination of the branching patterns of the sclerobiont colonies. The original settlement location of the colonial organism cannot be distinguished, but the branching pattern of colonies grow in a direction largely parallel to the direction of the aperture of the host.

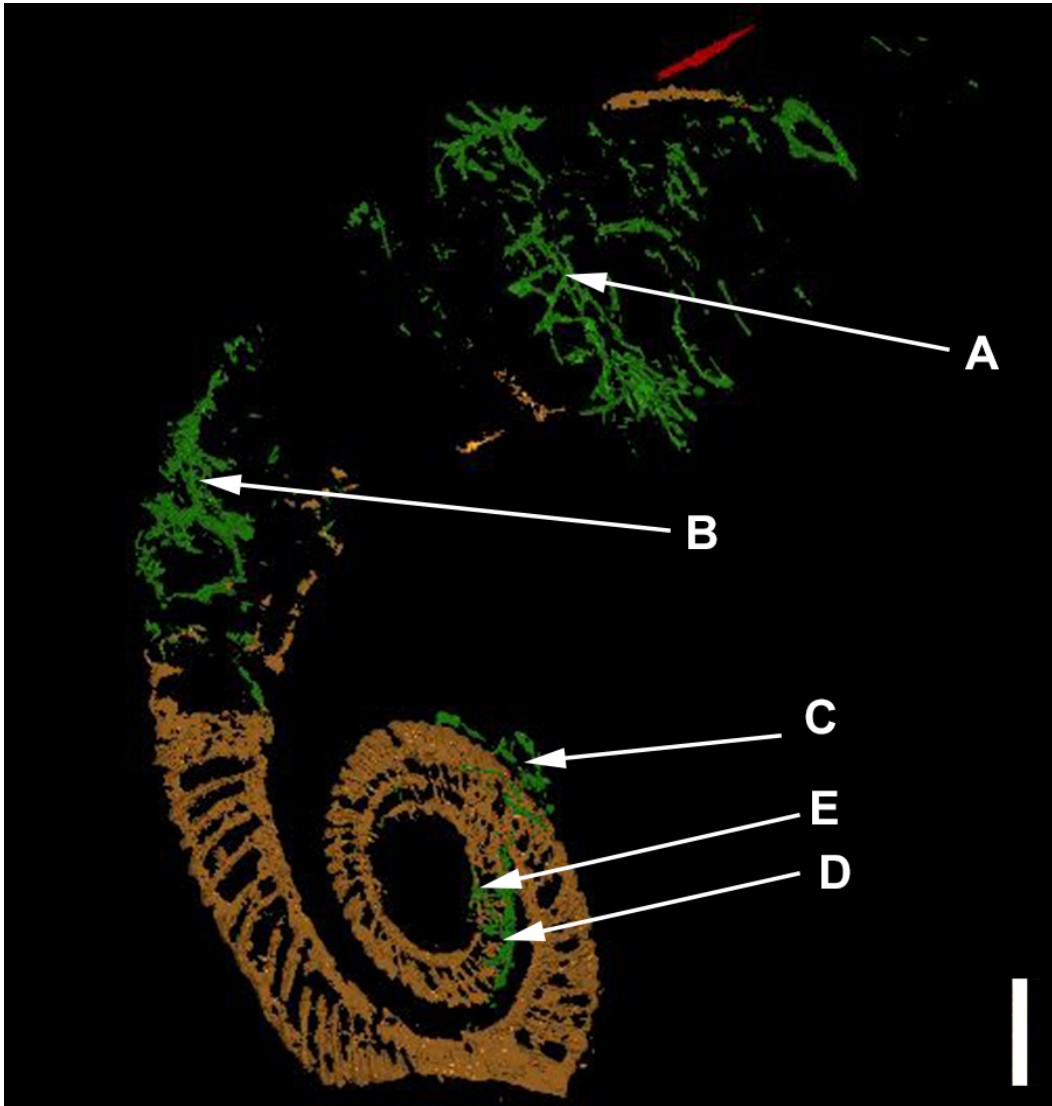

**Figure 3** **The ammonoid (brown), the epicoles (green) and the orthoconic nautiloid (red) in the 3D-model.** Five sclerobiont clusters (A–E) were distinguished. Scale bar: 1 cm.

The colonies in all clusters have the same taxonomic affinity. Zooids in the colonies are long and tubular, and curve slightly near the zooecial opening. Zooids are larger than those commonly observed in bryozoans and bud in alternating directions. The diameter of the more 3D-preserved tubes is typically around 1.5 mm, but this might have been artificially augmented by compaction (*De Baets et al., 2013b*). The tubes widen in the direction of growth expand distally and slightly contracted at the apex giving them a club-like appearance, characters typical of hederelloids (*Elias, 1944*).

Branching morphology in the established colonies on the host are diverse despite many shared characteristics. Morphologies of zooids exhibited in clusters A through D are generally more elongate and acutely curved away from the uniserial plane to those in cluster E which contain zooids that curve more dramatically along multiple, pluriserial

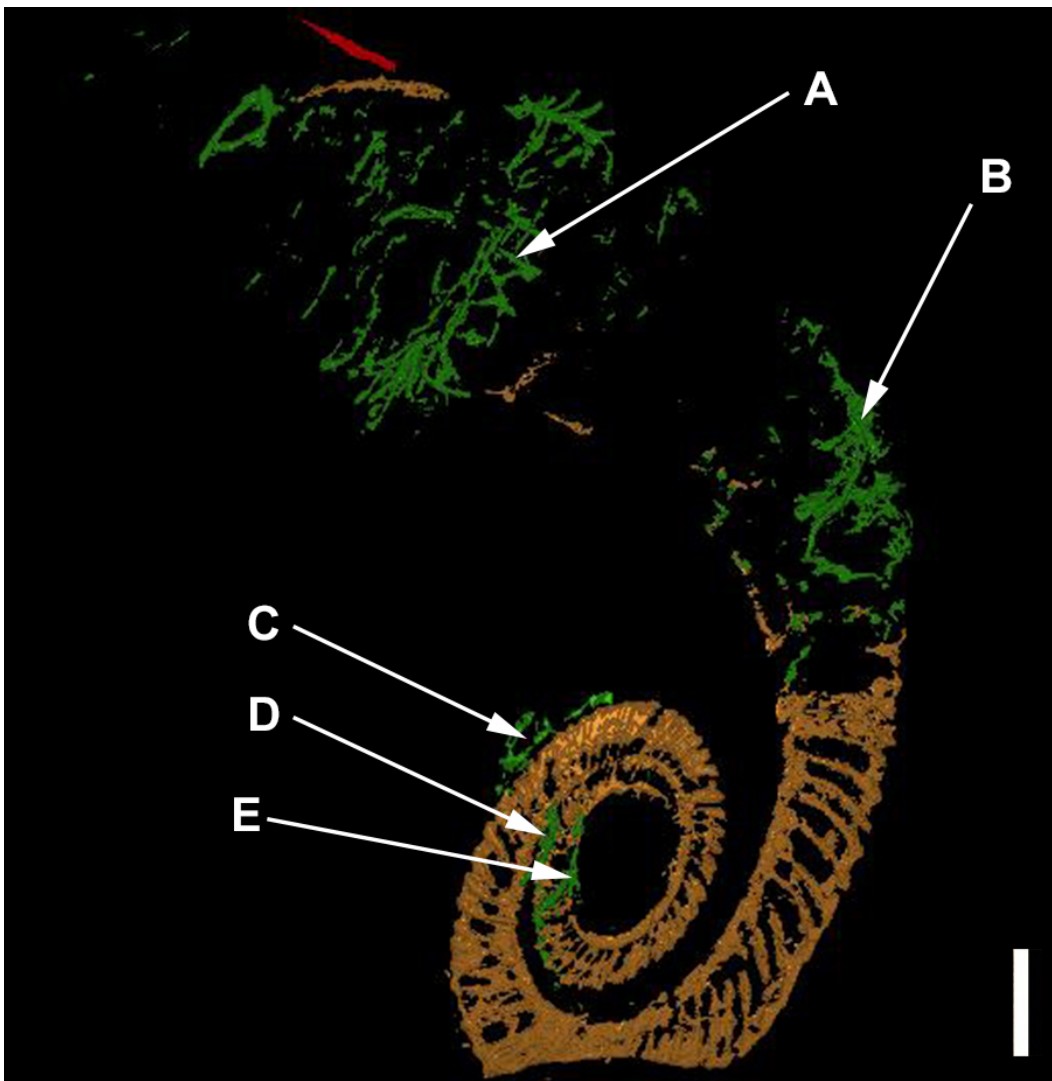

**Figure 4** **The ammonoid (brown), the runner-like epicoles (green) and the orthoconic nautiloid (red) rotated 180°.** Scale bar: 1 cm.

branches of the colony. Some of this variation could have been amplified by differential compaction in shales (*Ross, 1978*; *Briggs & Williams, 1981*).

## DISCUSSION

The three-dimensional model of *I. opitzi* permitted extensive observation of the relationship between the ammonoid host and the sclerobiont clusters present. The results permit positive identification of the sclerobionts and interpretation of the relationship between them and their host.

### Synvivo vs. post-mortem encrustation

Three possible scenarios can explain encrustation on both sides of the ammonoid by a sclerobiont colony, clearly visible in clusters C, D and E:

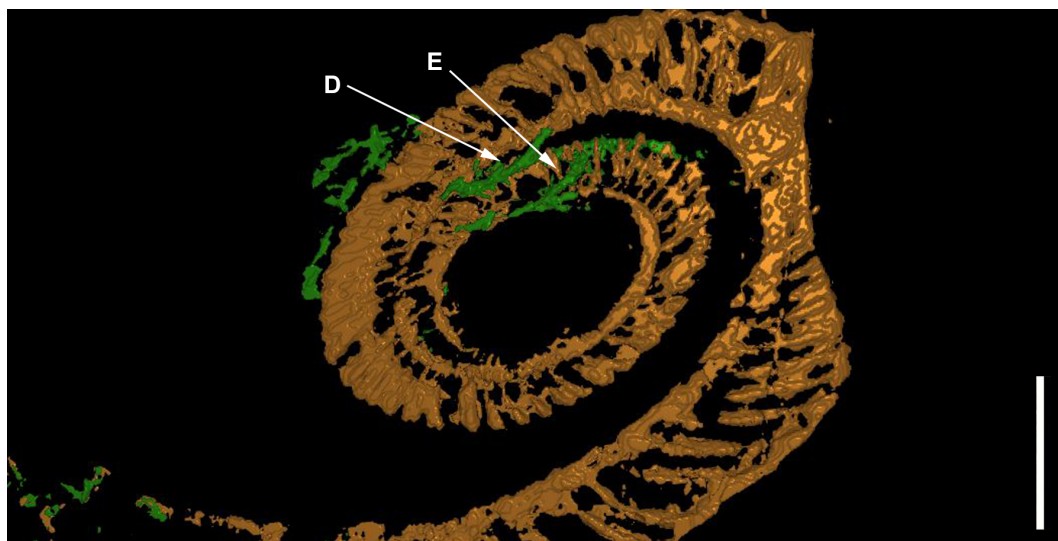

**Figure 5** **The white arrows mark the position of the sclerobiont clusters (D and E) close to the position where non-planispiral coiling can be first recognized.** Scale bar: 1 cm.

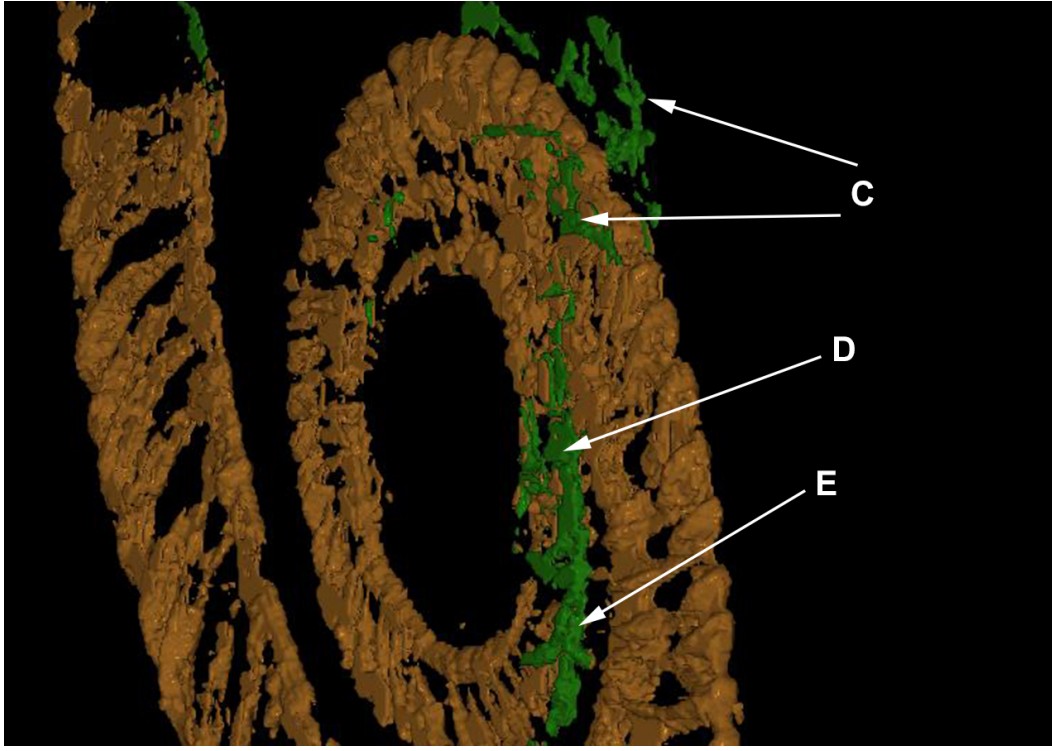

**Figure 6** **Close-up of the 3D-model showing that sclerobionts are settling on both sides of the shell.** Note correlation between asymmetrical sclerobiont encrustation of cluster E and the deviation from planispiral coiling of the host.

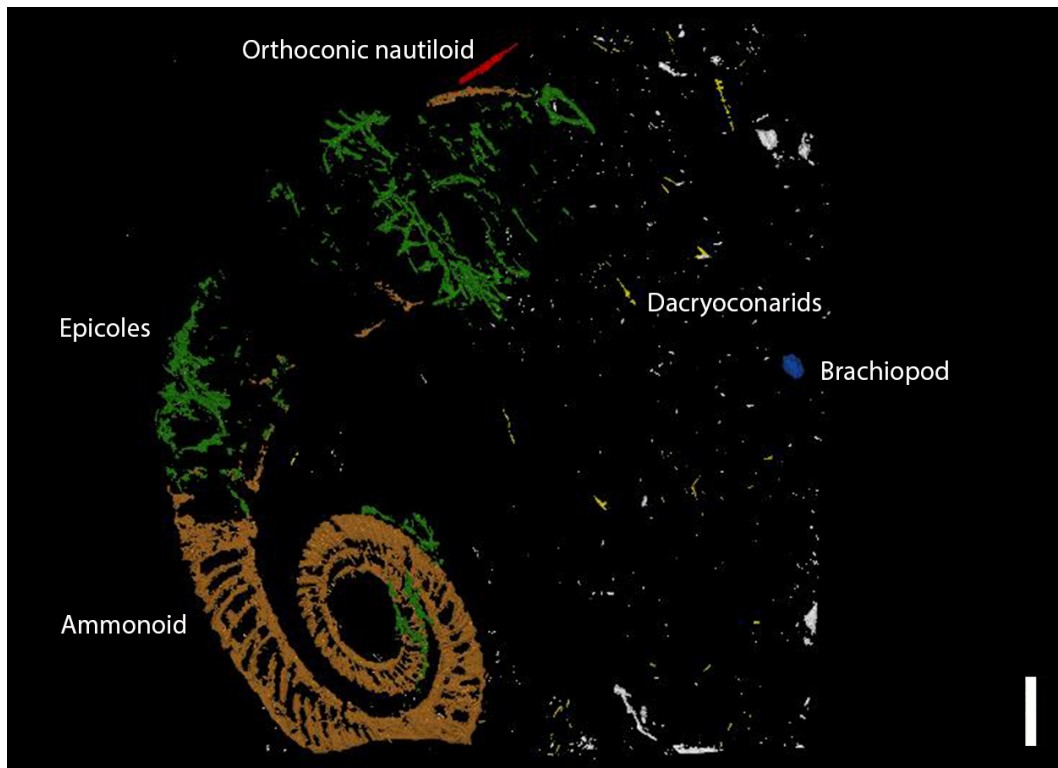

**Figure 7** **The 3D-model showing all components.** A brachiopod is colored blue and dacryoconarids are shown in yellow. Runner-like epicoles are marked in green and orthoconic nautiloid in red. Scale bar: 1 cm.

(1) *In vivo* encrustation: encrustation on both sides of the ammonoid by sclerobiont colonies C, D, and E and the direction of the growth of those sclerobionts matching the growth of the ammonoid, speak for an *in vivo* encrustation—at least for sclerobiont clusters C–E. A lack of a similar directional growth for all sclerobiont colonies on the ammonoid and with dacryoconarids in the surrounding substrate, as well as a lack in widespread encrustation across the ammonoid shell, further support the likelihood that the ammonoid was encrusted before the ammonoid settled on the seafloor (see Fig. 8).

(2) Post-mortem drift: encrustation of such large colonies solely during post-mortem drift seems unlikely due to the small size of our specimen (105 mm diameter). This is below the 200 mm limit listed for long floating cephalopod shells by *Wani et al. (2005)* and *Rakociński (2011)*. Interestingly, our specimen would have even a much smaller phragmocone volume than coiled ammonoids at equal diameters used in these experiments. More importantly, the extra weight of the sclerobionts should have made it sink even sooner than a non-encrusted shells so that it would not have resulted in encrustation on both sides. Post-mortem drift is deemed rare for ammonoids in general; most are implied to sink rather rapidly (*Maeda & Seilacher, 1996*). Furthermore, its vertical position in the water column should have been affected by asymmetric encrustation resulting in a non-vertical position of the shell after loss of soft-part parts which should have led to a more asymmetrical

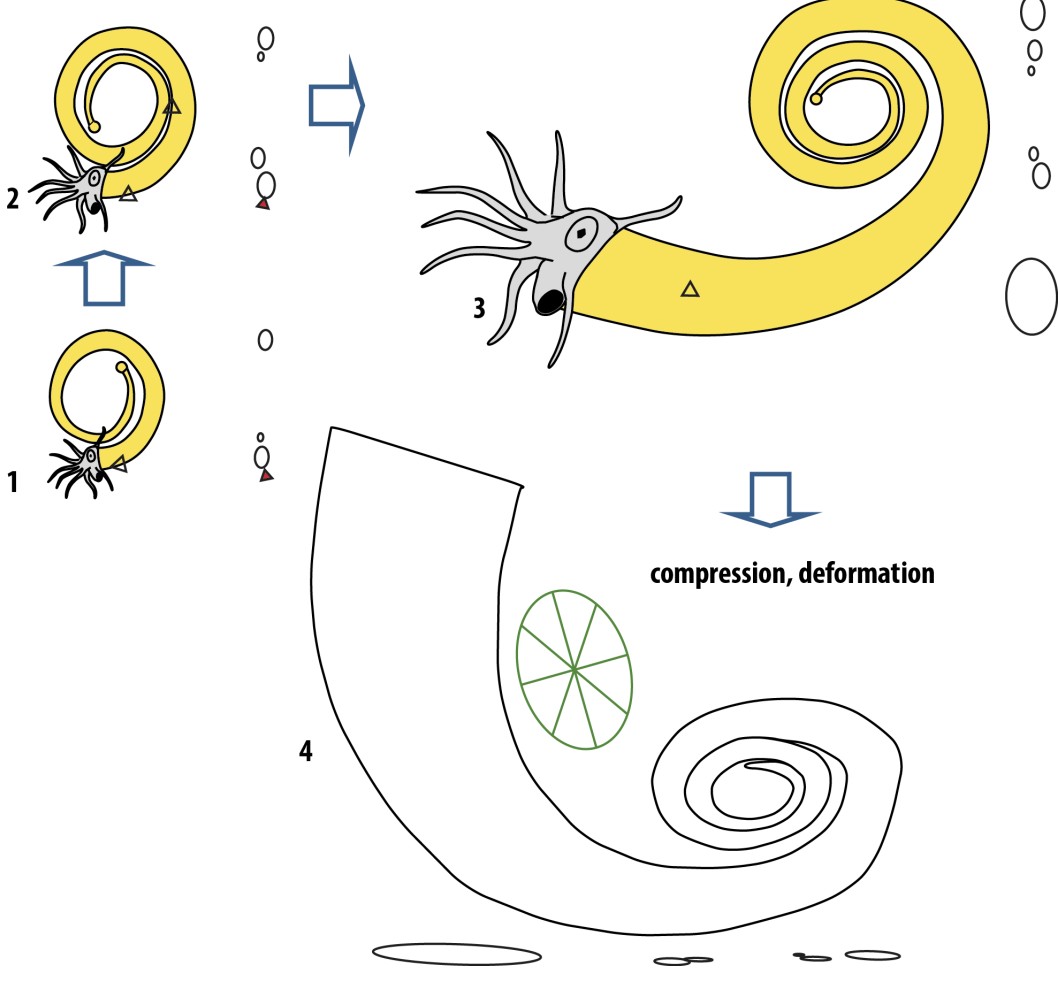

**Figure 8** **Model of *in vivo* encrustation and subsequent compression and deformation on *Ivoites opitzi.*** (1) orientation and conch morphology (in cross-section) of the specimen shortly after asymmetrical encrustation of first generation of epizoa (clusters D and E). (2) orientation and morphology of the specimen shortly after the asymmetrical encrustation of the second generation (cluster C) of epizoa (previous cluster of epizoa are currently lodged between the first and second whorl), (3) orientation and morphology of specimen when becoming encrusted with final epizoa (clusters A and B) or shortly before, (4) specimen after deformation; effect of deformation if specimen would have been initially planispirally coiled, effect in whorl section of specimen which was initially non-planispiral.

distribution of the sclerobionts as they preferentially attach to the lower part of drifting shells (*Donovan, 1989*).

(3) Resedimentation or reelaboration/reworking on the seafloor: usually encrustation of the sediment-free side of the ammonoids which mostly end up horizontal on the seafloor—is taken to be characteristic unless reelaboration/reworking of shells or internal moulds happened (*Macchioni, 2000*). In our cases, we have encrustation on both sides by the same colonies and transport by currents and reworking seems unlikely due the completeness of our specimen and similarities in preservation with other specimens of its taphonomic group (*De Baets et al., 2013b*). All their characteristics speak for a relatively rapid burial and infilling with pyrite during early diagenesis (before shell dissolution).

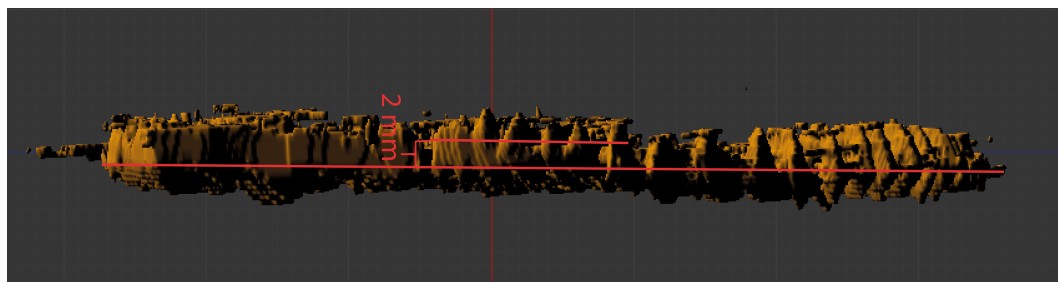

**Figure 9** Retrodeformed whorl cross-section to 200% results in a 2 mm distance between the midpoint of a particular part of the whorl (lateral view).

Furthermore, the same sclerobionts (C–E) started growing on the venter to both sides of the lateral sides which speak rather for a simple encrustation history rather than multiple generations of epibionts encrusting each sediment-free side sequentially (at least one before and after resedimentation/reworking and possibly more). No evidence for resedimentation or reworking (abrasional features) is present in any of the 82 studied specimens of *Ivoites*.

## Pathological variation in the morphology of *I. opitzi*

The non-planispiral coiling of this specimen, a unique occurrence in the species, occurs at the position of the sclerobiont clusters D and E. This pathological variation in host growth is also consistent with encrustation of the cephalopod during its life-time. Even if the deviation in coiling plane is only 1–2 mm now, it might have been substantially greater before burial and compaction. Whorl thickness alone has estimated to reduce up to 248% in some specimens (*De Baets et al., 2013b*), so originally these deviations could have been up at least 2.5–5 mm. If we artificially increase the thickness to 200%, we obtain a maximum deviation of the median plane of one whorl to the next of about 2 mm (2.5 mm if we augment the thickness to 250%). The same deviations can be observed within a single whorl (Fig. 9). Considering that the specimen has been extremely flattened (one whorl has been pressed on the other). At least another whorl thickness, the differences in whorl thickness between this whorl and the next which would make it a total of about 2 times this amount, ca. 4 mm (5 mm is we artificially augment the whorl thickness to 250%). This is a rather conservative estimate as we cannot know exactly how the specimen was compressed between the whorls.

Sclerobionts in clusters D and E are in a position that would have been hard to encrust if the subsequent whorl had already grown at the time of encrustation (e.g., when it was lying on the seafloor, the space between whorls around the venter would only have been between around 5 mm, complicating sclerobiont settling. Our specimen suggests that the second whorl lies on top of the first whorl (e.g., in the direction of the more heavily encrusted side, see Fig. 3) while the last whorl probably lies again below the second whorl (e.g., the coiling had almost normalized until the next encrustation by sclerobiont cluster C). The encrustation and its slightly different weight distribution across the venter would predict a deviation of the whorl initially in the direction of the encrusters weight and subsequently in the opposite direction (this seems to be still visible in our specimen despite it been heavily

flattened). Such a pattern would thus be expected if the specimen was encrusted *in vivo* for the first time slightly before it completed about one whorl (*Checa, Okamoto & Keupp, 2002*). The non-planispiral coiling observed in this specimen could not be produced by flattening or deformation—not even if the specimen was embedded obliquely (see discussion in Material and Methods). Furthermore, clusters D–E are positioned closely before the position where deviation of planispiral coiling can first be recognized and indicate that the coiling was induced by the sclerobiont encrustation. If these clusters grew on the ammonoid during life, as is also suggested by growth on both sides of the shell, the deviations from planispiral coiling would have been unavoidable if they are encrusted heavier on one side than on the other (*Checa, Okamoto & Keupp, 2002*).

The fact that sclerobiont clusters C, D, and E of runner-like epicoles are growing on both sides of the ammonoid and that clusters (D–E) occur slightly before the position where the deviations from planispiral coiling—where a whorl lies directly on top of the subsequent whorl—can be first recognized are also consistent with an encrustation of the ammonoid during its lifetime.

This specimen, therefore, documents the oldest direct evidence for *in vivo* encrustation of ammonoids. The previous record holders were *Paranarcestes*, *Latanarcestes* and *Sellanarcestes* from the Upper Emsian interpreted to be encrusted with auloporid corals during their lifetime, also evidenced by deviations from normal planispiral coiling in the host and by the subsequent growth of the ammonoid over the coral (*Klug & Korn, 2001*; *De Baets, Keupp & Klug, 2015*).

## Identity of the encrusters

Hederelloids are a problematic group of runner-like sclerobionts, which occur in the fossil record from the Silurian through Permian and are most diverse in the Devonian (*Solle, 1952*; *Solle, 1968*; *Taylor & Wilson, 2007*). Hederelloids have traditionally been treated as cyclostome bryozoans (*Bassler, 1939*; *Elias, 1944*; *Solle, 1952*; *Solle, 1968*; *Dzik, 1981*), but were redefined based upon differences in branching patterns, skeletal microstructure, lack of an astogenetic gradient, and wide range in tube diameters (*Bancroft, 1986*; *Wilson & Taylor, 2001*; *Taylor & Wilson, 2007*). They are currently mostly interpreted to be closely related to phoronids (*Taylor & Wilson, 2007*; *Taylor, Vinn & Wilson, 2010*; *Frey et al., 2014*). Both auloporid corals and hederelloids are uncommon in the middle Kaub Formation as their life habit requires a solid substrate upon which to settle; these were rare within the clay environments of the Hunsrück Slate (*Bartels, Briggs & Brassel, 1998*).

The nature of the pyritic preservation of the host specimen, *I. opitzi,* does not make it possible to look at the microstructure or fine details of the sclerobionts, but the general morphology supports that all clusters of sclerobionts share a taxonomic affinity. Runner-like sclerobionts common in the Devonian include auloporid coral, paleotubuliporid bryozoans, and hederelloids. The sclerobionts encrusting *I. opitzi* are colonial, with elongated zooids, lateral branching/budding patterns, and relatively large tube diameters which speak for their identification as hederelloids rather than auloporids or bryozoans (*Elias, 1944*; *Bancroft, 1986*; *Wilson & Taylor, 2006*; *Taylor & Wilson, 2007*).

The runner-like encrusters were initially thought to be auloporid corals (*De Baets et al., 2013b*), which are known to encrust brachiopods (*Zapalski, 2005*; *Mistiaen et al., 2012*) and ammonoids (*Klug & Korn, 2001*) during their lifetime. Some auloporid taxa have been confused with hederelloids in the past (*Fenton & Fenton, 1937*; *Elias, 1944*) and can be hard to differentiate when taphonomic conditions have degraded the quality of the specimen due to gross morphological similarities.

Hederelloids have been reported to encrust externally shelled cephalopods before (*Thayer, 1974*; *Brassel, 1977*; *Bartels, Briggs & Brassel, 1998*; *Frey et al., 2014*), but these are, to our knowledge, the first reported to encrust an ammonoid *in vivo*. Sclerobionts can provide also important information on paleoecology, sedimentary environments and taphonomy, both when they encrust shells *in vivo* or after death of their host (*Baird, Brett & Frey, 1989*; *Kacha & Šaric, 2009*; *Rakociński, 2011*; *Brett et al., 2012*; *Wilson & Taylor, 2013*; *Luci & Cichowolski, 2014*; *Wyse Jackson, Key & Coakley, 2014*; *Luci, Cichowolski & Aguirre-Urreta, 2016*).

The presence of five distinct clusters of hederelloids suggests that colonization of the host by sclerobionts happened numerous times (see Figs. 4 and 6). Not all colonizations show clear evidence of *in vivo* encrustation; clusters A and B were likely the last to settle on the specimen—based upon their location on the shell and the ontogeny of the host—but also do not show evidence of post-mortem encrustation. Clusters A and B, however, are the largest, which suggest that they had sufficient nutrients to settle, establish, and grow for an extended period of time. This is unlikely to have occurred in the benthos after the death of the *I. opitzi* specimen, but is not impossible. However, because the size of the hederelloid colonies in clusters A and B exceed that of the other colonies, clusters C, D and E were likely not living concurrently with clusters A and B.

## Implications for paleoenvironment of *Ivoites*

Some authors have attributed the rarity of encrusters in coiled ammonoids compared with Paleozoic (orthoconic) nautiloids to their ability to (keep) clean or use chemical defenses against encrusters (*Donovan, 1989*; *Davis, Klofak & Landman, 1999*; *Keupp, 2012*). Potentially, early ammonoid were more similar to their orthoconic nautiloid and bactritoid relatives. Furthermore, mechanical removal of epizoans might have proven more difficult in loosely coiled ammonoids like *Ivoites* as the previous whorl might have been out of reach of the cephalopod-arm complex (see Fig. 8). However, these suggestions are all quite; it is impossible to find direct evidence for chemical or mechanical cleaning or defensive behavior in this fossil group. More importantly, these mechanisms are also unnecessary to be invoked for such differences which could equally be explained by the fact that many nautiloids would have lived in more shallow environments—which show higher encrustation rates in general (*Brett et al., 2012*; *Smrecak & Brett, 2014*; *Smrecak, 2016*). Modern *Nautilus* shows differences in encrustations between different environments too (*Landman et al., 1987*).

The sedimentary environment of the Hunsrück Slate have been widely interpreted, ranging from shallow sedimentation on tidal flats, at depths from around storm-wave base to significantly deeper (*Solle, 1950*; *Seilacher & Hemleben, 1966*; *Erben, 1994*; *Bartels,*

*Briggs & Brassel, 1998*). The sedimentary environment in the central Hunsrück Basin is now thought to be rather complex; depressions between sedimentary fans provided environments below storm wave base and sills which could locally extend into the intertidal zone (*Etter, 2002*). The beds with the exceptionally preserved fossils and hemipelagic fauna (ammonoids, dacryoconarids) in the Bundenbach-Gemünden area have been interpreted to be mostly deposited just below storm-wave base to depths around 100 m (*Sutcliffe, Tibbs & Briggs, 2002*; *Stets & Schäfer, 2009*). Even in these regions, more sandy layers with neritic brachiopods are intercalated, suggesting occasional shallower depths above storm wave base. It is now generally accepted that the maximum depth was within the photic zone due to the presence in some layers of receptaculitid algae, which are interpreted to be closely related to green algae, and good visual capabilities of arthropods (*Bartels, Briggs & Brassel, 1998*; *Etter, 2002*; *Selden & Nudds, 2012*). Based on the latter, maximum depths of 200 m have suggested (*Rust et al., 2016*).

Hederelloids are typical encrusters found in Devonian photic zone environments (*Brett et al., 2012*). They are most diverse and abundant in shallower facies, but persist into the deep euphotic zone (*Smrecak, 2016*). The presence of hederelloids on the specimen, in combination with other evidence described earlier, support an interpretation of *in vivo* encrustation. The pelagic life habit of *I. opitzi* would allow colonies of hederelloids to settle and grow on the shell with some success. In contrast, at depths of 100+ meters, hederelloid encrustation, and sclerobiont encrustation in general, is significantly less common (e.g., *Brett et al., 2011*; *Brett et al., 2012*; *Smrecak & Brett, 2014*). Thus, presence of multiple colonies of hederelloids on the specimen support *in vivo* encrustation at depths within the photic zone, and lends further support for depositional conditions in line with current interpretations that the Hunsrück Slate interpreted was deposited near storm-wave base (*Bartels, Briggs & Brassel, 1998*; *Sutcliffe, Tibbs & Briggs, 2002*).

*De Baets et al. (2013b)* found encrusted ammonoid shells to be rare (only 6 of 342 studied ammonoids: about 2%). Only two specimens of 82 specimens of *Ivoites* (2%) were found to be encrusted with hederelloids; our specimen and an additional specimen of *Ivoites* sp. which was interpreted to be encrusted post-mortem (*Bartels, Briggs & Brassel, 1998*). Taphonomic or collection biases (*Wyse Jackson, Key & Coakley, 2014*) are unlikely to explain the low percentages of encrustation as the ammonites and epibionts are principally preserved in the way (e.g., pyritic compound moulds in our case). So far, bivalves, brachiopods, bryozoans, crinoids, hederelloids and tabulate corals have been reported to encrust conchs of ammonoids or other externally shelled cephalopods from the Hunsrück Slate s.s. or middle Kaub Formation (*Brassel, 1977*; *Bartels, Briggs & Brassel, 1998*; *Jahnke & Bartels, 2000*; *Kühl et al., 2012a*; *De Baets et al., 2013b*), but these have mostly thought to have happened post-mortem due to their heavy encrustation on one side of the fossils or the encrustation of the structures which would normally be covered with soft-parts (*Bartels, Briggs & Brassel, 1998*; *Jahnke & Bartels, 2000*; *De Baets et al., 2013b*). Heavy encrustations seem to be more common in nautiloids (*Bartels, Briggs & Brassel, 1998*; *Jahnke & Bartels, 2000*; *Kühl et al., 2012a*), but these have so far only been qualitatively studied. In environments between 100 and 200 m encrustation is generally low, which also consistent with a rare *in vivo* encrustation of our specimen which likely

swam in shallower depths when the first encrusters settled rather than the commonly reported post-mortem encrustation. Additional studies on epicoles on ammonoid shells and other shells from the Hunsrück Slate would be necessary to further corroborate these hypotheses.

## Implication for mode of life

Loosely coiled early ammonoids are mostly treated as poor swimmers based on their poor streamlining with high drag (*Westermann, 1996*; *Klug & Korn, 2004*; *Klug et al., 2015a*); additional limitations imposed by epizoa on streamlining and shell orientation might be (even) less important in these forms than in normally coiled ammonoids. The fact that our specimen survived at least three separate encrustations—as evidenced by different settlement locations of the clusters of hederelloids—and growth deformations associated with earlier encrustation further corroborates this idea, although further investigations on additional specimens would be necessary to confirm this hypothesis. Most hederelloid colonies generally grow along the spiral direction and do not cross from one whorl to the next, which provide additional evidence that they encrusted the ammonoid during its lifetime.

We cannot entirely rule out a post-mortem encrustation of clusters A–B. Hederelloid growth in those clusters is preferentially orientated away from the aperture of the ammonoid conch, as opposed to those of the inner whorls (cluster C–E) which are preferentially orientated towards it. Associated dacryoconarids do not show a preferential orientation with respect to the substrate (as would be expected in the case of current alignment: *Hladil, Čejchan & Beroušek, 1991*) or the hederelloids. This does not necessarily speak against encrustation during the lifetime of the ammonoid by clusters A and B as the terminal uncoiling of the ammonoid is interpreted to have influenced the life orientation from an upturned aperture in the inner whorls to a downturned aperture (see Fig. 8) during the terminal uncoiling at the end of the ontogeny (*Klug & Korn, 2004*; *De Baets et al., 2013b*; *Klug et al., 2015a*). We know the ammonoid specimen reached adulthood because it terminally uncoils, which is typical for many taxa of Anetoceratinae and interpreted as a sign of adulthood (*De Baets, Klug & Korn, 2009*; *De Baets et al., 2013b*; *De Baets, Klug & Monnet, 2013*; *Klug et al., 2015b*).

*Hederella* is known to encrust another ammonoid from the Hunsrück Slate, but this is interpreted to have happened post-mortem (*Brassel, 1977*; *Bartels, Briggs & Brassel, 1998*). There is no evidence that this happened *in vivo* in the second specimen as these are located on an incompletely preserved body chamber, and have been interpreted as encrusting the inside of the shell (*Bartels, Briggs & Brassel, 1998*)—but this should be further tested with μCT. Additional studies would be necessary to confirm if our specimen is an isolated case of *in vivo* encrustation or part of a more common phenomenon.

## Implications for taxonomy

Defining pathological specimens as species can have important taxonomic implications (*Spath, 1945*). According to *Spath (1945)*, such ammonoid species should remain valid, but no new type can be chosen while the holotype is still in existence. Others, like *Haas*

*(1946)*, claim assigning a pathological specimen might undermine the status of the species, considering the morphology of the species is described based on a pathological specimen, and that a new neotype should be selected. It is important to note that several authors have erected different ammonoid species based on small differences in coiling. As the only known non-planispirally coiled specimen of *I. opitzi* is likely pathological, an author like *Haas (1946)* might have suggested to designate another specimen as type for this species to avoid ambiguity. However, our study highlights that non-planispirality does not belong to normal intraspecific variation of this species, nor to the taxonomic definition of this taxon. Paratype SMF-HF 940, which was collected from the same locality as the holotype (*De Baets et al., 2013b*), would be the best candidate among the paratypes. Reassigning a neotype has recently been intensively discussed for the holotype of *Homo floresiensis* (*Kaifu et al., 2009*; *Eckhardt & Henneberg, 2010*), but the severity of deviation has to be considered in this specimen. However, both aspects (electing a neotype for pathological specimens and its dependence of the severity of the pathology) are not specifically discussed in the International Code of Zoological Nomenclature (ICZN). According to Article 75.1 of ICZN, "the neotype is … designated under conditions … when no name-bearing type specimen is believed to be extant …". In this case, if the holotype, even when pathologically deformed, is extant, the proposal of neotype is not granted. A proposal could be submitted to the ICZN to resolve the use of pathologically-induced morphological variation of holotypes, but this falls outside the aim of our study. Furthermore, we are confident that the original type specimen belongs to same species as the paratypes as it completes the same amount of whorl before uncoiling, has a similar rib spacing and only differs from other specimens in its minor coiling deviations (*De Baets et al., 2013b*). Such minor coiling deviations are not considered sufficient to erect a new species by us and other authors - even if these would not be of pathological nature (*Dietl, 1978*; *De Baets et al., 2013b*; *De Baets et al., 2015*). As non-planispiral coiling was also not part of the original diagnosis, we feel it is unnecessary to submit a proposal to appoint a new type specimen, which might not be allowed anyway.

## CONCLUSIONS

With the aid of μCT, we can demonstrate that at least some of the encrustations must have happened during the lifetime of the ammonoid as the sclerobionts are located on both sides of the ammonoid at the place where deviation from planispiral coiling starts. This indicates that the non-planispiral, slight trochospiral coiling in this specimen is probably pathological and does not form a part of the natural variation (*De Baets et al., 2013b*). To avoid taxonomic confusion resulting from non-spiral coiling in this taxon, which is not part of the natural variation as initially thought, one could select a neotype. While this practice has been suggested for other taxa (e.g., *Haas, 1946*), this is, in our opinion, not necessary as non-planispiral coiling did not form part of the original diagnosis. As the specimen survived at least 3 different encrustations and associated deformations through adulthood, the effects on its daily life were probably negligible. We re-identify these runner-like epizoa as hederelloids (as opposed to auloporid tabulate corals), which make them the first known hederelloids to encrust an ammonoid *in vivo* and suggests that the ammonoid probably

lived within the photic zone for most of its life. However, more studies on sclerobionts from the Hunsrück Slate, preferably with µCT, are necessary to further corroborate these hypotheses.

## ACKNOWLEDGEMENTS

JS performed the analysis in the framework of her Bachelor thesis (under supervision of KDB). Michael Wuttke, Markus Poschmann (Generaldirektion Kulturelles Erbe RLP, Koblenz) and Alexandra Bergmann (Steinmann Institute, Bonn) kindly borrowed and scanned the specimen for KDB in 2011. Peter Göddertz (Steinmann Institute, Bonn) kindly provided the original data and additional information on the original scan. We would also like to thank Andrej Ernst, Paul Taylor and Mark Wilson who kindly pointed us to the appropriate literature on hederelloids. The constructive reviews of the reviewers Ryoji Wani (*Yokohama*) and Russel Garwood (Manchester) are greatly appreciated. Trisha Smrecak also reviewed an early version of this manuscript. After the 1st round of revision, we invited Trisha Smrecak to join us as a co-author due to her extensive input and original additions to the research concerning the epizoa.

### Funding

The initial investigation of this specimen and scan was performed in the framework of the PhD of Kenneth De Baets, which was funded by the Swiss National Science Foundation (Projects 200021–113956/1 and 200020–25029 to Christian Klug). The funders had no role in study design, data collection and analysis, decision to publish, or preparation of the manuscript.

### Grant Disclosures

The following grant information was disclosed by the authors:
Swiss National Science Foundation: Projects 200021–113956/1 and 200020–25029 to Christian Klug).

### Competing Interests

Kenneth De Baets is an Academic Editor for PeerJ.

### Author Contributions

- Julia Stilkerich analyzed the data, wrote the paper, prepared figures and/or tables, reviewed drafts of the paper.
- Trisha A. Smrecak wrote the paper, reviewed drafts of the paper.
- Kenneth De Baets conceived and designed the experiments, contributed reagents/materials/analysis tools, wrote the paper, prepared figures and/or tables, reviewed drafts of the paper.

## Data Availability

The video with the digital visualisation of the investigated ammonoid Ivoites schindewolfi (KGM 1983/147), which is the data commonly uploaded in similar studies, and the primary data (images stack), have been supplied as Supplemental Files.

## Supplemental Information

Supplemental information for this article can be found online at http://dx.doi.org/10.7717/peerj.3526#supplemental-information.

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
