# Peer review of "D-Analysis of a non-planispiral ammonoid from the Hunsrück Slate: natural or pathological variation?"

_PeerJ, doi:10.7717/peerj.3526_

## Round 0.1 · original submission · Major Revisions

· Academic Editor

Major Revisions

Dear authors,

I have now three reviews of your manuscript, and all consider this study very interesting, well written and following the standards of PeerJ. However, in my opinion, as the Academic Editor for your manuscript, the anatomical, taxonomical and taphonomic problems that they detected are essential to make the manuscript acceptable for publication in PeerJ. Even, I consider that many of the concerns described by the reviewers are crucial for supporting the aim of this article. I recommend you take into account all the comments from the reviewers and especially attention should be given to the reviews from Dr. Ryoji Wani and Dr. Thrisha Smrecak (a typewritten copy of the main concerns of this last reviewer is provided below, to facilitate reading). Here are some of my main concerns: how you arrived to the conclusion that the studied specimen has non-planispiral coiling; comparing with what? Your interpretation is not well explained in the text or in the figures. I am also intrigued about why the change in the shell coiling can be a pathological trait that does not produce any apparent damage to the animal, rather to be a natural, slightly noted, intraspecific variation, as it was suggested for other species, or a post-mortem deformation. I would like also to know how many specimens belong to the species Ivoites opitzi to see if the apparently slightly trochospiral coiling that you describe for the holotype is significant. It is also important to know if epicoles are not observed in the shell of the other known specimens belonging to this species that have planispiral coiling. These are very important aspects in your manuscript, which have been also noted by reviewers, that need to be better developed.

Thus, please, revise carefully all the useful recommendations made by the reviewers or provide the reasons that you consider are enough evidence for refusing them.

Hoping to see your improved manuscript very soon!

Best regards,
Graciela Piñeiro

Most important concerns from Trisha Smrecak:
Interesting, but your thesis remained unclear until the conclusion. This is highly problematic.
You need to:
1-establish a case for how much deviation from typical the holotype is because of encrusters.
2-describe in a figure the difference in morphology caused by encrusters. You begin to do this well in “implications for taxonomy”.
3-better establish both “in vivo” argumenent and implications of the observed relationships to cause morphologic change
4-establish how rare non-planispiral coiling is in the taxon to add support for your desire to create a neotype.

Some lesser concerns:
A-Your other arguments (e.g. earliest noted live-live between hederellids ?ammonoids) have as little support for them as your thesis. I don’t disagree (but would like a better image of hederellids for confirmation). You haven’t supported them with sound arguments herein.
B-‘Pathological variation’ and ‘epicole’ are terms I would avoid if you want others interested in fossil parasitism, commensalism, encrusters to take note. Maybe ‘induced morphological variation’ and encruster/esclerobiont.
C-Your title and abstract don’t lead the readers to think that I. opitzi should have a neotype replacing the holotype. Rather, you lead me to believe that the species was named wrongly for a plastically deformed specimen of another species.
D- Your photic zone argument has validity but not as you wrote it. I don’t know as much as I should about the depositional unit, but you won’t find encrusters deeper than 250 mt. You almost never (less than 0.1%) in deep Cincinnati Arch environments (approx. 100 mt at most) and similar specimens exist in Middle Devonian App. Basin.

·

Basic reporting

I feel that this manuscript is well designed to PeerJ. English is very clear (at least I feel so), because this manuscript is easily understandable even for non-native speaker (=me). The manuscript has sufficient introduction and background to demonstrate. Figures are beautiful, which is relevant to the content of the manuscript. The raw data is shown in figures and supplementary file (3D images in this manuscript). Therefore, in this point of view, the manuscript totally follows the standards of PeerJ.

Experimental design

I feel that the experimental design of this manuscript is well organized. The investigation of the manuscript was conducted rigorously with a high technical standard. Therefore, in this point of view, the manuscript totally follows the standards of PeerJ.

Validity of the findings

I found several points that should be revised, to increase the robustness.
(1) non-planispiral ammonoid?
The authors mentioned that the examined specimen has non-planispiral coiling. However, I cannot totally understand why the authors state based on the completely flattened preservation. I guess that this point is self-evident for the authors and has been already discussed in the author’s previous studies (De Baets et al., 2013). However, this point should be clearly and fully mentioned and explained in the revised manuscript.

(2) How did the authors recognize the phragmocone and body chamber parts?
The authors distinguished the phragmocone and body chamber parts of the examined specimen. However, as far as I see the figures, I cannot understand why the authors can recognize the phragmocone and body chamber parts. This should be fully mentioned and explained in the revised manuscript.

(3) Result of Mophology of epicoles
The authors mentioned the morphology of the epicoles (lines 211-224). However, there is no citation of figures, so that it is very difficult to understand. For example, although the authors mentioned “the shorter axes” or “the branching patterns”, I cannot understand where is such characteristics in figures. I think the morphological description is very critical to identify the encrusters, so this point should be fully mentioned and explained in the revised manuscript, with appropriate citations of figures.

(4) Recognition of synvivo encrustation
The authors judged the encrustation occurred during ammonoid’s life, from the points of (1) epicoles are on both side of the specimen, and (2) epicoles are on the position where the deviation start planispiral coiling. But I feel that this author’s consideration is hasty. In the case of taphonomic encrustation, ammonoid shells are possibly encrusted on both sides. Therefore, I think that the authors should explained more in detail about point (2). However, this is not fully explained in the current manuscript. First, I cannot understand how we can recognize “the position that the deviation starts planispiral coiling”, based on the completely flattened specimen. Second, if the epicoles are on such position, I cannot understand why such epicoles indicate the encrustation during ammonoid’s life. In the current manuscript, the discussion of this topic is too short (lines 227-233). I think that lines 281-286 should be moved to the discussion of this topic. Such observation of morphology of epicoles is critical how the epicoles attached to the ammonoid shell.

(5) Minimal effect on mode of life?
In the Abstract and Conclusion, the authors mentioned that the encrustation had minimal effects on its mode of life. However, this point is not discussed in the manuscript (lines 273-298). As far as I think, if the encrustation occurred during the ammonoid’s life time and triggered the deviation from planispiral coiling (e.g., line 279), as the authors postulated, their mode of life must be changed. The transition from planispiral coiling to non-planispiral one would be critical change for ammonoids. Even so, if the authors want to mention “the minimal effect on mode of life”, this should be fully explained in the revised manuscript.

(6) Need neotype?
The authors mentioned that “neotype has to be chosen to define this species”, together with some references (Kaifu et al., 2009; Eckhardt & Henneberg, 2010; line 306). However, I think this treatment is inappropriate. The authors should cite International Code of Zoological Nomenclature, first. According to Article 75.1 of ICZN, “the neotype is … designated under conditions … when no name-bearing type specimen is believed to be extant …”. In this case, the holotype, even if the specimen is pathologicall deformed, is extant, so I believe the proposal of neotype is not admitted. If the authors still want to propose neotype, the authors should contact ICZN to get the admission, before describing the candidate specimen no. Current manuscript yields much confusion, without such procedure, and the description of neotype is totally incomplete (line 310).

(7) Several minor suggestions
1. Line 70: Please change to “Empty shells …”
2. Line 85, Bartels et al., 1998; Line 94, Bartels, Briggs & Brassel 1998; Which is appropriate to PeerJ style?
3. Line 114: Ruan (1996) and De Baets et al. (2010) are not in the reference list. Please add them.
4. Line 183: The authors mentioned that the epicoles are on dorsal side. But I think that the epicoles are also on the ventral side. Please describe more in detail with appropriate words.
5. Fig. 5: There are plural arrows in Fig. 5. I’m very confused why the plural arrows indicate one position that non-planispiral coiling can be first recognized. The line through the plural arrow’s points is the position? Or?
6. Line 270: Even if the previous studies mentioned the post-mortem encrustation, it is possible to discuss together with them. How are post-mortem encrustations? Concordant to the author’s suggestion that the epicoles of this study were attached in the photic zone (< 200 m in depth).

Additional comments

This manuscript is of great interest for many paleontologists, because it provides an interesting information about epicoles on ammonoid shell. However, I found some points that should be revised, to increase the robustness. Therefore, I would like to strongly recommend accepting this interesting manuscript for the publication in PeerJ, with minor revision.

·

Basic reporting

Basic reporting is good. I provide a number of minor suggestions on improving clarity of expression in the attached document.

Experimental design

This submission meets all of the Experimental Design points listed.

Validity of the findings

Please see my comments in the attached file - results are valid, but I recommend including a couple of lines/paragraphs to clarify just a couple of points.

Additional comments

Please see attached document. I look forward to seeing this in print.

·

Basic reporting

Please see notes on pdf.

Experimental design

please see notes on pdf

Validity of the findings

please see notes on pdf

Additional comments

please see notes on pdf.

---

## Round 0.2 · Minor Revisions

· Academic Editor

Minor Revisions

Dear authors,

I firstly want to apology for the delay on the decision for the manuscript “3D-Analysis of an early ammonoid…” submitted to PeerJ, but considering that one of the previous reviewers of the manuscript is now one of the authors in this new version (glad to know about that), and regarding that some of the main concerns from other reviewer were not revised as I expected, we (in accordance with the PeerJ staff) decided to ask for the opinion of a new reviewer. Unfortunately, the selected colleague did not answer in a reasonable time, and I understand that you need to know my decision. Well, I have to say that I am pleased to see that the manuscript has been interestingly improved, but, in my opinion as your academic editor for PeerJ, there are still some issues that should be addressed. These are important because they are crucial to define the objectives of your article.

1- The change from planispiral to non-planispiral coiling of the studied ammonoid conch is a condition that you interpreted because of the presence of the sclerobionts producing a deviation of 1 to 2 mm between the fragmacone whirls. But, that condition was impossible to achieve directly from the provided figures, for me, and in general it was a concern shared by all the reviewers. I think that for to solve this crucial aspect, you should provide a drawing or a 3D image that show the specimen in lateral view. For instance, Figure 6 of your new version, shows that the fragmacone is somewhat deviated at the level of putatively the last whirls respect to the rest of the conch. I cannot tell you how much deviated is, because the figure lacks a scale bar. Thus, it is necessary to include one in this and in all the figures that you provide. See for instance Figure 3 in Checa et al., 2002 which you cited, as a reference.

2- If the epizoans infected the individual during life, and set around the venter it is very possible, according to previous workers, that the shell grows to the opposite side where the epizoans were attached (see for instance Schmid-Röhl and Röhl, 2003). The case under study, as you stated, seems to show the contrary, as the aperture appears to be deviated to the side where the epizoans are attached. Maybe it is just a wrong perspective view, which can be solved by providing more evidence, as may be the lateral view of the specimen.

3- As you did not see notorious deformations in this specimen, it is also a possibility that the the epizoan halted its growth early and the ammonites continue growing at the normal planispiral coiling?. The following sentence extracted from your discussion could be indicative of the mentioned possibility?: “The encrustation and its slightly different weight distribution across the venter would predict a deviation of the whorl initially in the direction of the encrusters weight and subsequently in the opposite direction (this seems to be still visible in our specimen despite it been heavily flattened).”

4- It is important to show clearly the deviation of the shell growth from planispiral to non-planispiral coiling, as this specimen was colonized by epizoans, apparently in its lifetime. As you should know, despite deviations of ammonoid conch growth has been allied to epizoan infestation, not in all the cases the epibionts were preserved. Thus, if properly demonstrated, this could be one of the scarce cases that could prove that relation.

5- Environmental conclusions are interesting, but I think that it is important to include a discussion about the factors that led to colonization of living conchs of Ivoites by encrusters and why it was unable to avoid epizoan attachment. This could be another factor to strengthens your conclusions about epizoan attachment during the ammonite life. As you know, living ammonite colonization by epibionts is difficult to prove but according to some studies can be a non-rare phenomenon under certain environmental, limiting conditions. That discussion also should pay attention to the monotypic (?opportunistic) colonization found in Ivoites.

6- Your “Implications for Taxonomy” are based on current non-convincing evidence. The specimen is not a pathological individual, as one of the reviewers and I mentioned previously, it does not show signs to have been affected by the epizoans. If you can demonstrate that there is a very slightly deviation enough to interpret a non-planispiral coiling, that is a characteristic that is not the proper for the species but one that could appear depending on the eventual (very rare) colonization of the shell by epizoans. Thus, I recommend that this section should be removed from the manuscript.

Hoping that you will find the comments above interesting to strength the objectives of your manuscript, I very look forward to see the following version with the required changes.

With my best regards,
Graciela Piñeiro

---

## Round 0.3 · Minor Revisions

· Academic Editor

Minor Revisions

Dear authors,

I am glad to see the new submission of your manuscript “3D-Analysis of a non-planispiral ammonoid from the Hunsrück Slate: natural or pathological variation?” and how much it was improved, being particularly useful the additional figures. Despite I still think that there is no reason to consider the existence of a pathological behavior in the specimen that you are describing, I understand that there could be different interpretations and maybe your conclusions can be subject of discussion for other contributions in the future. I am willing to accept the publication of your article in PeerJ, but before you need to fix a few language issues that yet remain, along to references that were not cited in the text, all which were tracked and commented in the attached pdf. Given the suggested changes will not take much time, I hope to see the resubmission of the improved manuscript very soon.

With my best regards,
Graciela Piñeiro

---

## Round 0.4 · accepted · Accept

· Academic Editor

Accept

Dear Dr. De Baets,

I still found some minor details that should be addressed before acceptation. Please find them as tracked changes in the attached pdf; I have accepted all the previous changes to make easier the detection of the few new suggested changes. However, as they are just minor grammatical matter, I believe that you can edit them once you receive the final proofs of your article.

Congratulations!

Sincerely,
Graciela Piñeiro